# Influences of Maternal Weight and Geographic Factors on Offspring Traits of the Edible Dormouse in the NE of the Iberian Peninsula

**DOI:** 10.3390/life13051223

**Published:** 2023-05-21

**Authors:** Silvia Míguez, Ignasi Torre, Antoni Arrizabalaga, Lídia Freixas

**Affiliations:** 1Independent Researcher, E-08016 Barcelona, Spain; 2BiBio Research Group, Natural Sciences Museum of Granollers, C/Francesc Macià 51, E-08402 Granollers, Spain; itorre@mcng.cat (I.T.);

**Keywords:** *Glis glis*, Catalonia, reproduction, offspring number and size trade-off, maternal effects, litter size, offspring body weight, geographical gradients

## Abstract

The main goal of this study was to analyze the reproductive patterns of edible dormouse (*Glis glis*) populations in the northeast of the Iberian Peninsula using an 18-year period of data obtained from nest boxes collected between 2004 and 2021. The average litter size in Catalonia (Spain) was 5.5 ± 1.60 (range: 2–9, n = 131), with litter sizes between 5 and 7 pups as the more frequent. The overall mean weight in pink, grey and open eyes pups was 4.8 g/pup, 11.7 g/pup and 23.6 g/pup, respectively. No differences in offspring weights between sexes were found in any of the three age groups. Maternal body weight was positively associated with mean pup weight, whereas no correlation between the weight of the mothers and litter size was found. The trade-off between offspring number and size was not detected at birth. Regarding litter size variation across the geographic gradient (and their climatic gradient associated) from the southernmost populations of the Iberian Peninsula located in Catalonia to the Pyrenees region in Andorra, no evidence to suggest that geographic variables affect litter size was found, discarding (1) an investment in larger litters to compensate shorter seasons related to higher altitudes or northern latitudes, and (2) variation in litter size related to weather changes (e.g., temperature and precipitation) along latitudinal and/or altitudinal gradients.

## 1. Introduction

The life history theory seeks to understand the life cycle and describes how natural selection, as the principal underlying force, acts to optimize life history traits that favor allocating organisms’ strategies to produce adaptations in response to the environment [1,2,3]. Organisms are forced to optimally trade-off limited time and energetic resources between life history traits to maximize fitness because two traits competing for the same resources cannot be maximized simultaneously [1,2]. Therefore, a trade-off occurs when an increase in fitness due to a change in one trait is opposed by a decrease in fitness due to a concomitant change in the second trait [4]. The most prominent life history trade-off involves the cost of reproduction [5]. Reproduction is energetically costly [6,7]. The trade-off between offspring number and size has been documented considerably in numerous studies, constituting a key point to many hypotheses and theoretical models of optimal litter size within life history research [8,9,10]. This negative relationship between number and size (or body mass used as a proxy of body size) of offspring in mammals, including rodents, has been reported both at birth [11,12,13] and at the end of maternal dependence or at weaning [11,12].

Several environmental and individual variables affect resource acquisition and allocation, which can impact the cost of reproduction [14]. Maternal effects can be defined as the causal influence of genotypes or phenotypes of mothers on the traits expressed by their offspring [15,16] and can impact offspring fitness, being important as adaptations to environmental stress and changing environment [17,18,19,20]. Among the maternal phenotype (non-genetic maternal effects) that influence offspring phenotypes (e.g., number and size of pups), the physical condition and body weight of mothers are two of the most meaningful traits: heavier females invest more in those offspring and/or produce heavier pups at birth than females in poor condition [21,22,23]. Many studies supported the positive relationship between maternal body mass and offspring or litter mass [11,24,25,26] and/or the number of offspring produced [11,27,28], both at birth and at weaning. However, other works did not report relationships between female body mass and litter size or offspring mass [25,29].

Key events in the organisms’ life depend not only on the trade-offs or physiology of an individual, but also on the environmental variation [30]. Resource availability (especially food availability) is one of the most important environmental factors affecting reproduction, because, depending on the quantity of food available, animals decide how much to invest in reproduction [31,32]. Furthermore, reproduction may be regulated and/or influenced by other environmental factors, such as temperature [33,34,35], rainfall patterns [35,36,37] and day length (photoperiod) [33,38]. These environmental factors are subject to temporal and spatial variations (e.g., between years or across latitude, longitude and altitude), with important implications on organisms’ life histories [39,40,41]. Geographic variations in reproductive characteristics, such as litter size, have been documented in a variety of vertebrates. Some studies reported that litter size increased with latitude and altitude [41,42,43,44], while others documented a decline [45] or simply showed no relationship [44,45]. With respect to the longitude, in certain works, no relationship with litter size was found [41].

The number of mammae and nipples varies greatly among groups of mammals, ranging from 2 to 29 [46]. Not only traits such as mother’s weight, seasonality, resource availability or environmental features are correlated with the number of offspring per litter [11,32,47,48], but also mammae and nipples number can explain part of the variation in litter size [48,49]. Specifically, the number of mammae and nipples is usually positively correlated with litter size [49]. Rodent species have, on average, one-half as many offspring as they have nipples [49].

The edible dormouse *Glis glis* (Linnaeus, 1766) is a nocturnal, arboreal and hibernating rodent that weighs approximately 100 g and inhabits deciduous forests with a distributional range covering most of Europe, the Caucasus and Asia [50,51,52]. Litter size ranges between 1 and 12 pups, commonly 5–8 [53,54,55,56,57]. This small mammal is a specialized seed predator that must cope with high annual variability in tree seed production [58] and represents an example of extreme adaptation to an unpredictable environment, such as fluctuating food [58,59]. Accordingly, this rodent uses a variety of strategies to respond to environmental fluctuations, such as hibernation, estivation, anticipatory reproduction and reproduction skipping, as well as short torpor and huddling, which are not uncommon adaptations among endotherms. However, their combined occurrence in a single species is unusual [51]. Total reproduction failures can be observed coinciding with the lack of seed mast in autumn [54,59,60,61]. Dormice seem to anticipate the future food availability (i.e., beech seeds and acorns), and they invest in reproduction (both sexes) in the current year only if food and seeds later in the year will be abundant to allow fattening of the juveniles prior to their first hibernation [62]. Although several studies showed that higher body mass does not increase reproduction or reproductive decision in the edible dormouse [58,59,60,63], less well known is the role of maternal effects, such as the influence of maternal body mass (as an indicator of body condition and nutritional status of the mothers), on offspring phenotype (particularly on offspring number and size) when the decision to invest in reproduction has already been made. Understanding the role it plays during the intermediate mast years (i.e., years in which only a part of the tree population produces seeds and a lower proportion of individuals reproduces) may be relevant to a better understanding of life history strategies in this small mammal. Concretely, some studies have investigated relationships among young body mass, litter size and maternal body mass, mainly at weaning, and others have also used food-supplementation experiments [53,64]. Overall, these relationships have been much less studied using the offspring at birth and focusing on the relevant role that food availability in spring, rather than autumn, could have on the mother’s body condition during pregnancy and not only during lactation. Unfavorable weather conditions during mating and pregnancy (e.g., cold and rainy periods) could be responsible for reproductive failure in the edible dormouse, but some findings did not show clear evidence of temperature and precipitation effects on low reproductive rates [65]. Other studies obtained similar results, in which reproductive decisions were not influenced by low temperatures and high precipitation in summer [59]. However, these arguments did not fully explain whether temperature or precipitation could affect, for example, litter size, when the decision to reproduce has been taken.

Climate change is a critical factor that clearly affects biodiversity [66,67]. Furthermore, several studies focused on the effects of climate change suggested that changing environmental conditions can produce changes in life history traits in numerous species [68,69]. Therefore, it is of particular interest to improve knowledge about the life history of the edible dormouse and how climate change could affect their reproductive patterns.

The main goal of this study was to analyze the reproductive patterns of edible dormouse populations living in Catalonia (Spain). For that purpose, the first part of this study describes in detail two important life history traits: litter size and offspring weight based on data obtained in the framework of the Dormouse monitoring program in Catalonia [70]. The second aim addresses questions related to reproductive energy allocation and life history strategies. In this sense, this study seeks to answer (1) whether the trade-off between the number and size of offspring can be detected in the edible dormouse at birth, (2) whether maternal body weight has an effect on offspring weight and litter size using the pups nearest in age to birth, and (3) whether litter size is a life history trait subject to geographic variation, using latitude, longitude and elevation as proxies of changes in environmental conditions, within a gradient from the southernmost populations of the Iberian Peninsula located in Catalonia to the Pyrenees region in Andorra.

## 2. Materials and Methods

### 2.1. Study Area and Sampling Design

This study analyzes reproduction data collected from the Dormouse monitoring program in the northeastern corner of the Iberian Peninsula [71,72], mainly in Catalonia (Spain) which started in 2004, and a minority of the data was compiled in Andorra, which started in 2008. This area lies between the eastern half of the Pyrenees to the north, the Mediterranean coast to the south-east and the driest areas of the Ebro basin to the west, with relief features that rise from sea level to more than 3000 m a.s.l. in the Pyrenees [73]. Annual rainfall ranges from 400 mm to more than 1500 mm, and the mean annual temperature is from 18 °C (on the southern coast) to less than 3 °C (in the alpine belt) [73]. In the specific case of Catalonia, it can be observed that according to the Environmental Stratification of Europe, it contains 4 of the 13 main Environmental Zones, and this makes it a highly diverse area rich in a variety of environmental conditions and landscapes [74,75]. The sampling sites were located in the following natural areas: (1) Montnegre i Corredor Park (with minimum and maximum nest boxes altitudes ranging from 460 to 764 m a.s.l.), (2) Montseny Natural Park-Biosphere Reserve (elevation range from 1014 to 1240 m a.s.l.), (3) Guilleries-Savassona Natural Area (range: 409–1047 m a.s.l.), (4) Capçaleres del Ter i del Freser Natural Park (range: 1309–1367 m a.s.l.), (5) Cadí-Moixeró Natural Park (range: 1092–1105 m a.s.l.), some site in (6) Vall d’Aran (range: 831–1709 m a.s.l.), and (7) Andorra (range: 1054–1858 m a.s.l.) (Figure 1). In Catalonia, some sampling stations were placed in monospecific beech forests (*Fagus sylvatica*). However, most of the study areas were located in mixed deciduous forests consisting mainly of sessile oak (*Quercus petraea*), accompanied by beech (*Fagus sylvatica*) and other deciduous trees such as maple (*Acer opalus*), chestnut tree (*Castanea sativa)*, whitebeam (*Sorbus area*), wild cherry (*Prunus avium*) and ash (*Fraxinus* sp.), *Tilia* sp. In addition, other oak species are also represented (e.g., holm oak (*Quercus ilex*) mainly in areas of strong influence of the Mediterranean forest (Montnegre and Montseny Massifs), Algerian oak (*Quercus canariensis*) concretely located in the Montnegre massif, or pubescent oak (*Quercus pubescens*)). The undergrowth was predominantly dominated by common box (*Buxus sempervirens*) in Pyrenean regions such as Capçaleres del Ter i del Freser Natural Park, or mainly composed of hazel (*Corylus avellana*), holly (*Ilex aquifolium*), common hawthorn (*Crataegus monogyna*) among others in more southern regions. Regarding the overall characteristics of the habitat according to the dominant trees and shrubs in Andorra, the sampling sites were located (a) in mixed deciduous forests dominated by sessile oak (*Quercus petraea*) and accompanied by other trees species such as ash (*Fraxinus* sp.), wild cherry (*Prunus avium*) and walnut (*Juglans regia*), and (b) in deciduous riparian forests with poplar (*Populus* sp.), birch (*Betula* sp.) and scotch pine (*Pinus sylvestris*). In the sampling stations placed at higher altitudes (>1600 m), mountain pine (*Pinus mugo* subsp. *uncinata*) was also present. The undergrowth consisted mainly of shrubs such as hazel (*Corylus avellana*), alpenrose (*Rhododendron ferrugineum*) and alpine juniper (*Juniperus communis* subsp. *nana*). Data collection was carried out following the protocols outlined on the website of the Dormouse Project (www.lirons.org, accessed on 15 May 2022), using nest boxes (size: 30 × 15 × 15 cm; diameter of entrance hole of 5 cm, at a height of approximately 2.5–3 m above the ground). The nest boxes were spaced at 25–30 m intervals, following two sampling techniques (line transect method and plot method) with a permanent location. The plot sampling method consisted of a 5 × 4 grid (n = 20 nest boxes; grid = 1 ha), while the transect method involved lines of approximately 150 m (n = 6 nest boxes). In both cases, the sampling stations (lines or plots) were spaced 250–300 m to ensure data independence (i.e., no interchange of individuals between sites). Data availability for this study included an 18-year period (2004–2021). However, the collection method, sampling effort, and sampling years differed among some of the study sites due to the impossibility of maintaining such a high sampling effort in all places during this long period (see Table 1 for details of the sampling design in each natural area).

The study of the edible dormouse populations was carried out using the mark–recapture technique. Individuals captured in nest boxes of both sexes and all ages (except pups) were marked with ear tags (Style 1005-1, National Band and Tag Co., Newport, KY, USA) and a subcutaneous microchip (AVID Musicc, 8 × 2.1 mm). Dormice were sexed, aged (adult, yearling, juvenile or pup), weighed with digital balance (±0.1 g) and for all individuals, their reproductive status was established. Furthermore, three biometric characters were measured using a vernier caliper: tail, hind foot and tibia lengths. Offspring were only sexed, aged and weighed. In order to standardize the data, three age categories were used for the pups based on their morphology: pink pups (from birth until grey color differentiation begins to appear in the dorsal area), grey pups (from grey colored pups to pups beginning to open their eyes) and pups with open eyes (from pups with fully open eyes and ear canals to pups with approximately 40 g and/or near to weaning). According to Vekhnik (2022) [76], it was estimated that pink pups were between 1 and 8 days old, grey pups between 9 and 21 days, and pups with open eyes between 22 and 35 days. All individuals born within a year were considered pups only when they had a pre-dispersal weight (<41 g) [54], while weights >41 g were recorded as a juvenile and were not considered in the analysis because they were already weaned.

### 2.2. Statistical Analysis

Data analysis was summarized following three blocks: (1) mean litter size, body weight of pups by age group and differences in offspring weight by sex and age, (2) trade-off between offspring number and size and effect of maternal body weight on mean pup weight, and (3) geographic patterns in litter size.

#### 2.2.1. Mean Litter Size, Body Weight of Pups by Age Group and Differences in Offspring Weight by Sex and Age

Mean litter size (i.e., LS, the number of pups born per litter) was described from 131 litters collected along Catalonia (NE Spain). Only data related to the first two age groups (i.e., pink pups and grey pups) were considered, owing to the increased risk of predation with the age of the pups [76]. Regarding offspring weight, 1170 data were used to describe the overall mean weight according to age groups (pink pups, grey pups and open eyes pups). Using a subset of data from the initial dataset (n = 1092), Mann–Whitney U tests (function *wilcox.test* in the package *stats* [77]) were performed for each age and sex group to examine any differences between pup weights because data did not follow a normal distribution (see Appendix A for more details).

#### 2.2.2. Trade-Off between Offspring Number and Size and Effect of Maternal Body Weight on Mean Pup Weight

The effect of litter size (i.e., LS, the number of pups born per litter) and maternal body weight (i.e., MBW, the weight of mothers with pink pups) on mean pup weight at birth (i.e., mPW, the average weight of a pup per litter) was investigated, as well as the correlation between LS and MBW. Only the age class nearest to birth (i.e., pink pups) was considered, which reduced bias due to weight change associated with lactation and also minimized distortions in the number of pups caused by mortality in older age classes [76,78]. For females, postpartum weight was used as a proxy for pre-pregnancy and/or gestational weight. The low sample size (n = 36) did not allow running linear mixed models, and therefore, correlation analyses and Ordinary Least Squares (OLS) linear regressions (using *cor.test* function and *lm* function in the *stats* package, respectively [77]) were performed. A set of candidate models was created with mPW as the response variable, and LS and MBW as explanatory variables (for additional information, see Appendix A).

#### 2.2.3. Geographic Patterns in Litter Size

Geographic patterns in litter size (LS) along latitudinal, longitudinal and altitudinal gradients were examined from data obtained in each nest box with the presence of pups, both in plots and lines. Geographic coordinates of nest boxes and elevation information were determined by means of the Global Positioning System (GPS). The data subset used for this analysis included litter size data from natural areas located in Catalonia (which provided most of the data) and Andorra. The area evaluated after data cleaning ranged between latitudes and longitudes of 41.659°–42.817° N and 0.669°–2.592° E, respectively, and an altitudinal range between 460 and 1497 m a.s.l. To test the relationship between geographic variables and the number of pups per litter in the edible dormouse, and considering that the response variable is a count data, a Generalized Linear Mixed Model (GLMM) with Poisson distribution and log link function [79] was initially built using the *glmmTMB* function in the *glmmTMB* package [80]. Latitude, longitude and elevation were the explanatory variables, and litter size was the response variable. ‘Sampling station’, ‘Nest box identity’, ‘Female identity’ and ‘Year’ were included as crossed random effects. To remove the collinearity problems detected between predictors, Principal Component Analysis (PCA) was carried out to extract the principal components (PCs). The singular value decomposition (SVD) was the method used to perform PCA using the *prcomp* function in the *stats* package [77]. The Kaiser criterion (eigenvalues >1) [81] was used to select the number of PCs to retain, resulting in only the first principal component (PC1). A second model was then built following the steps described above, with the PC1 scores selected under Kaiser’s rule being the new explanatory variable. The PC1 scores should provide a gradient of environmental conditions at small and mid-scale using the geographic variation as a proxy of environmental variation. Because data show less variation than could be expected based on a Poisson distribution, the model was refitted using the Conway-Maxwell-Poisson and Generalized Poisson distributions with a log link function to avoid problems with under-dispersed data (see Appendix A for details).

All statistical analyses were conducted in R version 4.0.5. [77]. Plots were generated using *ggplot2* [82], *ggpubr* [83], *ggridges* [84], *GGally* [85] and *cowplot* [86] packages. Mean, standard deviation (SD), and range were the descriptive statistics used to summarize and describe the variables of interest. To measure the strength and direction of the relationship between variables, Pearson’s (r) or Spearman’s (r_s_) correlation coefficients were calculated for quantitative variables, depending on the variables evaluated. The Variance Inflation Factor (VIF) and Tolerance indices were calculated for all fitted models and used in combination with the correlation coefficients to detect collinearity problems between explanatory variables using the *check_collinearity* function in the *performance* package [87] and *cor.test* function from *stats* package [77], respectively. Collinearity was considered to be present (1) when the absolute correlation coefficient |r| between explanatory variables was greater than 0.7 [88] and (2) when the variance inflation factor (VIF) was greater than 5 to 10 and the tolerance values were less than 0.1 to 0.2 [89]. Interaction effects with research interest were tested but removed from the set of candidate models if they were not significant before implementing a model selection method. Model selection was performed using Akaike’s information criterion corrected for small sample sizes (AICc). The model with the lowest AICc value and ΔAICc less than 2 was considered the model with the best fit [90]. The subset of candidate models was obtained using the *model.sel* function in the *MuMIn* package, and they were ranked according to their AICc [91]. All models with a ΔAICc < 2 were considered equivalent in relation to the model with the lowest AICc (best model) [90]. In this case, the final fitted model was obtained by means of a model averaging process in order to capture the overall effects of variables and to account for model uncertainty. Then, all regression coefficients in the models that were equally supported (i.e., within 2 AICc units) were averaged using the *model.avg* function in the *MuMIn* package [91]. Furthermore, the adjusted R^2^ values of the best models (R^2^_adj_) were also used as a measure of the goodness-of-fit for linear regressions. Alternatively, for the Generalized Linear Mixed Models (GLMMs), the variance explained by the best model was estimated by means of the pseudo-R^2^ values, concretely through the marginal R^2^ values (R^2^_m_), that is, the variance explained by the fixed effects [92] using the *r2_nakagawa* function in the *performance* package [87]. The conditional R^2^ value (R^2^_c_) was not reported because the variance of the random effects was close to zero. Even so, the random effects structure based on the experimental design was maintained. Model assumptions for the linear regressions were verified by visual inspection of the residual diagnostic plots using the *check_model* function in the *performance* package [87], as well as by performing Shapiro–Wilk normality tests, Breusch–Pagan tests for homogeneity of variance and Durbin–Watson test for autocorrelation in the residuals. In the case of GLMMs, the model fit assessment was based on residual diagnostics using the *simulateResiduals* function in the *DHARMa* package [93]. A *p*-value < 0.05 was considered statistically significant; a *p*-value ≤ 0.10 was mentioned as a tendency to an effect.

## 3. Results

### 3.1. Mean Litter Size, Body Weight of Pups by Age Group and Differences in Offspring Body Weight by Sex and Age

In Catalonia, the litter size of the edible dormouse showed a mean of 5.5 ± 1.60 (SD, range 2–9) pups per litter. For the 131 litters observed during the study period (2004–2021), 3.8% were litters of two pups (n = 5), 6.9% were litters of three pups (n = 9), and 15.3% (n = 20), 20.6% (n = 27), 23.7% (n = 31), 19.1% (n = 25), 9.2% (n = 12), and 1.5% (n = 2) of the litters had four, five, six, seven, eight and nine pups, respectively (Figure 2). No litter with one pup or with ten or more pups were found. The most frequent litter size was six, and a total of eighty-three litters (63.4%) had between five and seven pups.

A total of 1170 pups of edible dormouse were weighted. The overall mean body weight in the pink pups was 4.8 g/pup ± 1.13 SD and varied between 2.5 and 6.8 g (n = 140). The second age group (grey pups) ranged from 4.0 to 25.5 g, with a mean of 11.7 ± 3.88 g/pup (n = 579). Finally, the average weight for the open eyes pups was 23.6 ± 6.16 g/pup and ranged from 10.1 to 39.8 (n = 451) (Figure 3 and Appendix A).

The mean body weight of pink pups was 4.8 ± 1.02 g (range = 2.7–6.8 g; n = 51) in females and 4.9 ± 1.16 (range = 2.5–6.5 g; n = 75) in males. In grey pups, females also showed a slightly lower mean body weight than males, 11.2 ± 3.70 g (range = 4.0–24.7 g; n = 238) and 11.9 ± 3.92 g (range = 5.5–25.5 g; n = 299), respectively. Finally, in open eyes pups, females showed, on average, a body weight of 23.5 ± 6.41 g (range = 10.1–39.8 g; n = 195) compared with a mean weight in males of 23.2 ± 5.72 g (range = 12.6–38.5 g; n = 234). No statistically significant differences in offspring weight were found between females and males in any age group (Mann–Whitney U tests: *p* > 0.05; Figure 4, Appendix A), although the *p*-value for grey pups was marginally significant.

### 3.2. Trade-Off between Offspring Number and Size and Effect of Maternal Body Weight on Mean Pup Weight

The results showed that for the Catalan edible dormouse population, the mean postpartum body weight for 36 females with pink pups (offspring between 1 and 8 days of life) was 103.1 ± 13.65 g (range: 59.3 g to 132.9 g). The correlation between MBW and LS evaluated when the pups were pink was low and not significant (r_s_ = 0.15; *p* = 0.3898). In addition, a preliminary multiple regression analysis with mPW as a response variable showed that the interaction between these two variables was also non-significant. These results suggest that in the edible dormouse, litter size and maternal body weight were not associated, and furthermore, the effect of maternal weight on average pup weight was independent of the number of pups per litter (Appendix A). However, mean pup weight (mPW) and litter size showed a marginal correlation (r_s_ = 0.31, *p* = 0.0654). On the other hand, MBW after parturition was positively associated with mPW, with a significant correlation (r = 0.43, *p* = 0.0098) (Figure 5). From the ranked models, the two best models with good support (ΔAICc < 2) were selected (Table 2). The top model (lowest AICc) included only MBW as an explanatory variable (R^2^_adj_ = 0.156, F_(1,34)_ = 7.486, *p* = 0.00981), and the second-best model involved the main effects of the two predictors (LS and MBW) (R^2^_adj_ = 0.164, F_(2,33)_ = 4.434, *p* = 0.0197). The range for adjusted R^2^ showed that the two linear regressions explained between 15.6% and 16.4% of the variance, respectively. The final model (obtained by means of model averaging) revealed that litter size did not have an effect on mean pup weight (*p* = 0.62). Therefore, the trade-off between offspring number and size at birth was not clearly detected in the edible dormouse considering pink pups. On the other hand, the mean pup weight of pink pups increased with increasing maternal weight (*p* = 0.01, Table 3).

### 3.3. Geographic Patterns in Litter Size

A total of 135 litters collected along Catalonia and Andorra were used in this analysis. The PCA showed that 98.2% of the variability was explained by the first two principal components (PC1-PC2). According to the Kaiser criterion, only the first component was considered (eigenvalue = 2.14), which explained 71.5% of the total variance and was positively associated with latitude and negatively with longitude (Appendix A). The GLMM fitted using generalized Poisson distribution was clearly the top model and was selected as the most parsimonious (Table 2). The residuals were normally distributed, and the dispersion value was 1.07 (*p* = 0.6), which means that no over- or under-dispersion was detected, resulting in a satisfactory residuals check for the final model (Appendix A). According to the GLMM results, PC1 scores showed a non-significant effect on litter size (R^2^_m_ = 0.013, *p* = 0.172; Table 3), suggesting that the geographic variation assessed (and climatic variation associated with the latitudinal and altitudinal gradients) does not affect litter size.

## 4. Discussion

This study analyzed the reproductive patterns of edible dormouse (*Glis glis*) populations in the northeast of the Iberian Peninsula (Spain and Andorra) using an 18-year period of data obtained from nest boxes collected between 2004 and 2021. The results provide the first overall description of litter size and the offspring weight in a large territory of Catalonia (Spain), not only focusing the study on two specific natural parks as was carried out in previous work [94]. The average litter size (based on pink and grey pups) in Catalonia was 5.5 ± 1.60 (range: 2–9, n = 131), with litter sizes between five and seven pups more frequent. This result was similar to other mean litter sizes reported in European studies: an average of 5.8 in Slovenia [95]; 5.34 in the southern Alps (Italy) [59]; 5.9 in Lithuania [56]; and 5.4 in Austria [96], and was also similar to the averages of 5.21 and 5.92 documented in the Montseny and Montnegre massifs, respectively, the two southernmost edible dormouse populations located in Catalonia [94]. The findings are also in accordance with the literature regarding litter of 5–8 young are the most frequent in other European regions [55]. However, the value obtained was considerably lower than the 6.8 found in England [54] or the litter size of 7.85 estimated in Iran [97]. Litter size can vary depending on the number of studied litters, forage conditions of a locality in certain years or mammary number [62,97], among other causes. In the particular case of the Iranian population, the large difference could also be due to the different methodology used, because individuals were captured with kill traps, and litter size was estimated from embryos and placental scars [97]. Regarding the body weight of the offspring, some publications documented that the newborns weigh 2 g at birth [55], while other studies provided average birth weights of 3.4 g [98] and 3.7 g [99]. The average value for pink pups obtained in this study (4.8 g; all pups less than 8 days old) is moderately higher than those reported by other researchers for newborns, and is mainly due to the impossibility of obtaining data in the field where all litters have pups on the first day after birth. Even so, the minimum value of the range (2.5 g) was in consonance with the other European estimates provided. The weight of the pups depends on their development which can be affected by environmental factors. In fact, daily body weight gains of pups were quite variable in the literature: mean of 1.14 g [55] or 1.28 g within the first 30 days of life [99], and 1.9 g per day among the first weeks [54]. In the Istranca Mountains of Turkish Thrace, it was documented that pups reached a body weight of 20 g on day 26 of their lives [55], which in this study corresponds to the age group called “open eyes pups”. Then, the average of 23.6 g obtained in open-eyed pups was similar to those reported in other European regions. Several reasons could justify the higher dispersion observed in the weights of grey and open eyes pups compared to the homogeneous weight distribution obtained in pink pups. For instance, it may be due to the fact that the three ages differ in the range of days that they remain in each group, or also due to the influence of maternal weight. Furthermore, it is important to emphasize that the variation in offspring weight within a litter may be due not only to constraints on the maternal ability to allocate resources evenly among their offspring caused by environmental or physiological constraints [100,101], but may also be due to a strategy called diversified bed-hedging [101]. This strategy consists of maximizing the number of offspring of variable weights with the aim of maximizing fitness when environments vary temporally [100,101]. No differences in offspring weights between sexes were found in any of the three age groups considered, although they were not expected because previous findings documented an overall non-significant difference in morphometric traits (including body weight) between sexes in this species [60,102,103]. In this vein, the use of standardized age categories based on external characters [76] that could be independent of the weight—which can be affected by environmental factors affecting pup development—could be advisable for comparative purposes in monitoring schemes.

The second main goal of this study was (1) to investigate the trade-off between offspring number and size (using offspring weight as a proxy of body size), and (2) to evaluate the effect of maternal weight (used as a proxy of maternal condition or nutritional quality) on litter size and offspring weight in pink pups (age group nearest to birth). Contrary to expectations of a negative relationship between offspring number and size according to life history theory [2,3], the results revealed the non-existence of a trade-off between offspring mass and offspring number. Furthermore, no correlation was found between maternal weight and litter size. However, as expected, the mother’s weight had a positive effect on the mean offspring’s weight. Gestation and lactation involve significant energy expenditures [6]. Mothers adjust their energy allocation strategies based on environmental conditions that, in turn, determine the quality of available resources, which can affect the nutritional status and/or body condition of mothers, as well as the phenotype of offspring and juvenile survival (e.g., food restriction during pregnancy reduce litter size, pup body weight and pup developmental growth rates) [104,105,106,107,108,109,110]. Heavier females produce heavier offspring with better survival due to the overall positive relationship between juvenile survival and birth weight [111,112]. In addition, differences in resource acquisition (i.e., some individuals having access to more resources than others) may mask life history trade-offs [113]. Accordingly, in some cases, females can vary the strength of trade-offs among life history traits, affecting the life history favored by selection, which can compensate for the costs of reproduction [114]. This can produce positive correlations between traits rather than negative correlations expected under the theory [113,115,116]. In the edible dormouse, a positive effect of maternal body mass and a negative effect of litter size on the mean body mass of young at weaning were found. However, the statistical significance of predictors varied among years (e.g., litter size negatively affects pup mass at weaning, but other factors such as birth date or body mass of the mother may obscure this relationship in particular habitats or years) [53]. The positive association between maternal body weight and offspring weight obtained in this study indicates that this relationship can be detected not only at weaning, but also at the age nearest to birth (i.e., heavier mothers producing pups with higher birth weight), with interesting repercussions to have a better understanding of reproductive success in this rodent. Possibly, this positive effect is the main reason why the negative relationship between offspring number and size at birth was not found, because individual heterogeneity and environmental conditions can affect the strength of the trade-offs and, therefore, can mask their evidence [113]. Another cause could be related to the use of average weight per litter. Consequently, the existence of a trade-off between offspring size and number at birth cannot be rejected in the edible dormouse. In opposition, the relationship between maternal body weight and litter size seems to be more ambiguous in this small mammal. Some food supplementation experiments have shown that more food availability did not affect female body mass gain during gestation, and daily female body mass increase during gestation did not correlate significantly with litter size [64]. Other studies also obtained similar results, showing that supplemental feeding had no effect on female body mass [63], and neither reproductive activity nor litter size was significantly affected by pre-reproductive body mass [58]. However, a positive effect of the body mass of reproductive females on litter size at emergence from hibernation was observed [64]. According to this study, no correlation between litter size and maternal body weight was obtained, although an increase in the number of pups per litter could be expected when the body size of the mother increases, as occurs in other rodent species [11]. One reason could be related to the impossibility of obtaining field data where all litters contain newborn pups (e.g., the first day of life). However, the problem of bias caused by litter sizes that do not represent the real number of pups at birth [63] was considerably reduced in this analysis due to the use of the age group nearest to birth (litters with pups less than 8 days old). Secondly, previous research in other species also showed no correlation between maternal body mass and litter size [25,117]. Finally, considering that the positive effect of female body condition on litter size in the edible dormouse was established at the time of emergence, but in contrast, no effect was described using female body mass during gestation [64], these findings may also be caused by the use of maternal body weight just after parturition rather than female body weight at emergence from hibernation. After emergence from hibernation and during the spring and early summer, the edible dormouse consumes significant amounts of low-energy foods such as vegetative parts of plants, wild cherry, wild apple, raspberries and blackberries, among others, and also oak acorns from the previous year and some food of animal origin (birds, their eggs and insects and/or arachnids) [118,119,120]. In addition, during the same period of months (prior to parturition), the presence of flowers and the consumption of seed buds such as beechnuts or acorns (i.e., early stages of high-quality and caloric seeds) are an environmental signal to predict the autumn availability of these energy-rich seeds, which will coincide with lactation and therefore, dormice would use it as an anticipatory cue to trigger sexual capacity, but not as a source of energy needed for reproduction because body mass (as an indicator of body condition) is not a determining factor in the reproductive decisions of this small mammal [51,58,59,60,63,121,122]. Lack of reproduction in edible dormice is common to occur coinciding with the lack of food resources in autumn (i.e., non-mast years), where it can be observed that males remain reproductively inactive (absence of testicular growth) and lactating females are not found [54,60,62,65]. The main hypothesis explaining the reproductive investments of the edible dormouse suggests that at emergence from hibernation (both in beech and oak forests), individuals of both sexes will initiate investment in gonadal function and reproduction only if food resources with high energy and nutrient content are abundant, and females will maintain this investment in reproduction only if high-quality food continues to be present, otherwise, reproduction is aborted by resorption of embryos [51]. It is well known that under unfavorable environmental conditions, most females resorb embryos [123] or occasionally may eat their pups after parturition if food is not available [51]. As a whole, the findings of this study could suggest that years in which reproduction is triggered by the presence of inflorescences or seed buds with high caloric content (e.g., acorns and beechnuts), the availability of low-quality foods in spring and summer (foods that differ from those used as an anticipatory signal to predict the autumnal mast situation) could influence maternal nutritional status and/or maternal body mass during the mating period and especially during gestation, which in turn will determine offspring quality (concretely, offspring birth weight but not litter size). The size of the offspring at birth is related to their probability of surviving the lactation period [11]. Thus, the findings of this research support previous studies that documented a positive effect of maternal body mass on offspring mass at birth in mammalian species, including rodents [11,24,25,124,125]. The results are also in consonance with the importance that fruits may have in spring and summer since, in other regions, it was found that the presence of trees producing edible fruits such as different berries, plum trees, grapes, common figs or walnut affect edible dormouse habitat use [126]. In this rodent, these findings may be of particular relevance over the intermediate mast years (i.e., years in which only a part of the tree population produces seeds, in which only a fraction of females have litters) [58]. Given the preliminary nature of this study, further research is needed to corroborate the degree of importance of this hypothesis, as well as to understand the relevance of maternal effects on offspring traits at birth in this small mammal. Variance explained in mean pup weight only accounted for 15.6% and 16.4%, suggesting that many other untested maternal effects that can also affect offspring characteristics such as birth mass or litter size (e.g., maternal age) may also be influencing [125,127,128]. Although age affects reproduction in the edible dormouse, yearlings have smaller litters than adult females [58,63]. In this study, there were insufficient data from secure yearling females (i.e., marked as young and recaptured the following year), and therefore, it was not possible to test the effect of age.

Finally, a possible geographic variation in litter size along a latitudinal, longitudinal and elevational gradient (used as a proxy of climate gradient) was explored from the southernmost populations of the Iberian Peninsula located in Catalonia to the Pyrenees region in Andorra. No evidence was found to suggest that geographic variables affect litter size. Latitudinal and altitudinal geographic gradients provide environmental changes in factors such as temperature, precipitation, solar radiation and season length, among others [40,129]. Overall, at higher altitudes and latitudes, temperature decreases, and precipitation tends to increase, and these changes can have important effects on populations (e.g., on morphological and reproductive traits) [39,40,130,131,132]. Photoperiod also varies in relation to latitude, and different strategies can be observed to regulate reproduction seasonally accordingly to day length [33]. Thus, environmental factors can clearly influence mammalian reproductive parameters in several ways [33,35]. Traditionally, two general approaches have been used to explain latitudinal and altitudinal variation in litter size: (1) at higher latitudes, litter size increases in “non-hibernating prey species” to compensate for higher adult mortality rates in winter caused by relatively extreme environments associated with higher latitudes [42], and (2) altitudinal and latitudinal variation are related to the season length and parental mortality associated with reproduction in small mammals (including rodents), so short seasons limit the maximum number of times a female can reproduce in their lifetime, giving an advantage to phenotypes that produce large litters [43]. Other studies have also failed to find variations in litter size related to latitudinal and/or altitudinal gradients [44,45,133]. In the particular case of hibernators, relationships between litter size and environmental conditions such as snow depth, air temperature and precipitation were documented in marmots [134,135], and a positive relationship was found between mean spring temperature and litter size in the edible dormouse [57]. A maximum longevity of 14 years has been documented in the edible dormouse [136]. The energy investment in larger litters to compensate for shorter seasons associated with higher altitudes or northern latitudes, although it involves a decrease in the life expectancy of females, could be advantageous in small mammals with lower longevity than dormice. However, this reproductive strategy does not seem necessary in dormice, not only due to their longevity, but also because of the large number of strategies that this species shows (hibernation, estivation, anticipatory reproduction, and skip reproduction) [51]. Despite being a life history trait with large variation in animals, litter size does not seem to be influenced by the edible dormouse’s geographic gradient (or the climatic gradient associated). Consequently, during mast years (i.e., years with high-quality food available in autumn and in which reproduction is triggered in spring), resource availability is one of the key factors that could explain a considerable part of the variation in litter size in this rodent, according to other findings in which number of pups per litter increases with food availability [58,64]. This approach is also in agreement with other reports in which other reproductive decisions, such as the proportion of females that reproduce each year, were only correlated with seed production but not with temperature or rainfall [59]. Nevertheless, further and more specific studies about the effect of climatic variables (especially during gestation) should be conducted to elucidate whether temperature or precipitation could have an impact on litter size at birth in the edible dormouse. Although some studies used more restricted latitudinal and longitudinal ranges than those used in this study when the topography and climatology of the sampling areas varied markedly [137,138], it would also be of particular interest to assess variation in litter size along a larger latitudinal and longitudinal gradient. This additional research is also considered important in the context of climate change because it is necessary to know how climatic variables can influence life history traits and thus understand the effects of climate change on hibernating mammals and inform conservation planning [139].

## 5. Conclusions

In the edible dormouse (*Glis glis*) population located in Catalonia (Spain), the overall mean litter size found (5.5 ± 1.60 SD; range: 2–9) was similar to some other mean litter sizes reported in European studies. This study also provided an accurate description of offspring body weight at three ages (pink, grey and open eyes pups), giving the mean weight and range by age and sex. No differences in offspring weights between sexes were found in any of the age groups considered, corroborating the lack of sexual differences in body mass found in other studies.

Maternal postpartum body weight had a positive effect on mean pup weight in the age group nearest to birth (i.e., pink pups), suggesting that heavier mothers would produce pups with higher weight at birth. However, mean pup weight was not inversely related to litter size, as could be expected under a life history trade-off between offspring size and number, probably due to the positive effect of maternal weight on offspring weight that can be masking this trade-off. Accordingly, the trade-off between offspring size and number at birth cannot be rejected in the edible dormouse. Furthermore, no positive correlation was found between litter size and maternal body weight, although litter size would be expected to increase as maternal body size increases, as occurs in other species. These findings could suggest that the availability of low-quality food in spring and summer could influence maternal nutritional status and/or maternal body mass during the mating and gestation periods, affecting offspring birth weight but not litter size, with particular relevance in the intermediate mast years.

Finally, the analysis did not reveal geographic patterns (i.e., latitude, longitude and elevation) in litter size along a geographic gradient from the southernmost populations of the Iberian Peninsula (Catalonia) to the Pyrenees (Andorra), discarding (1) the reproductive strategy in which an energy investment in larger litters compensates for shorter seasons related to higher altitudes or northern latitudes, and (2) variation in litter size related to climatic changes (e.g., temperature or precipitation) along latitudinal and/or altitudinal gradients.

This study concluded that further research is needed to expand the understanding of the effect of maternal body weight and the influence of climatic variables (e.g., temperature and precipitation) on offspring traits such as pup weight and litter size at birth in the edible dormouse, and the implications this could have in the context of climate change for this hibernating mammal.

## Figures and Tables

**Figure 1 life-13-01223-f001:**
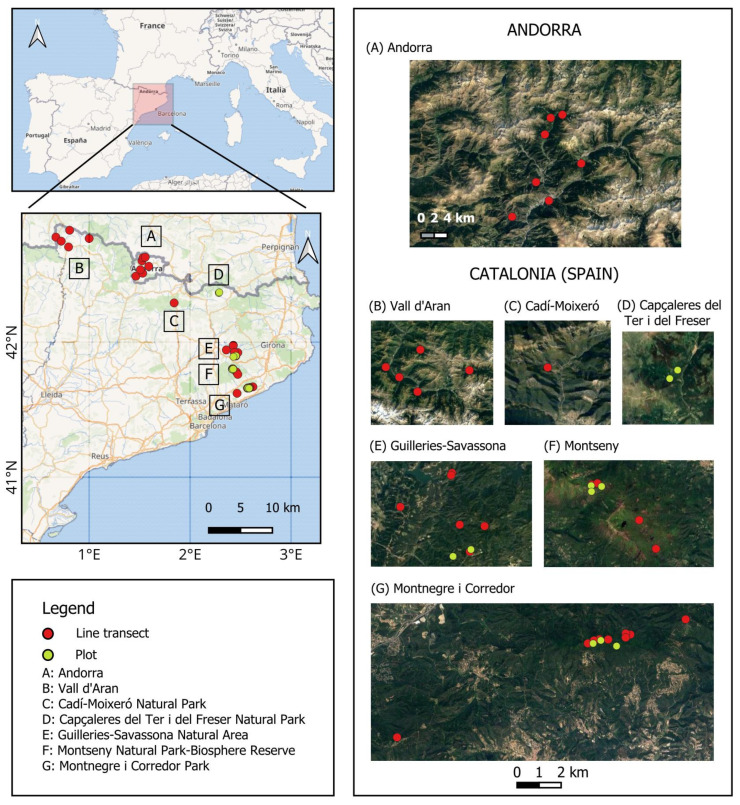
Map of the study area showing the location of all sampling stations in the NE of the Iberian Peninsula: Catalonia (NE Spain) and Andorra. Red circles indicate line transects, and yellow circles indicate plots.

**Figure 2 life-13-01223-f002:**
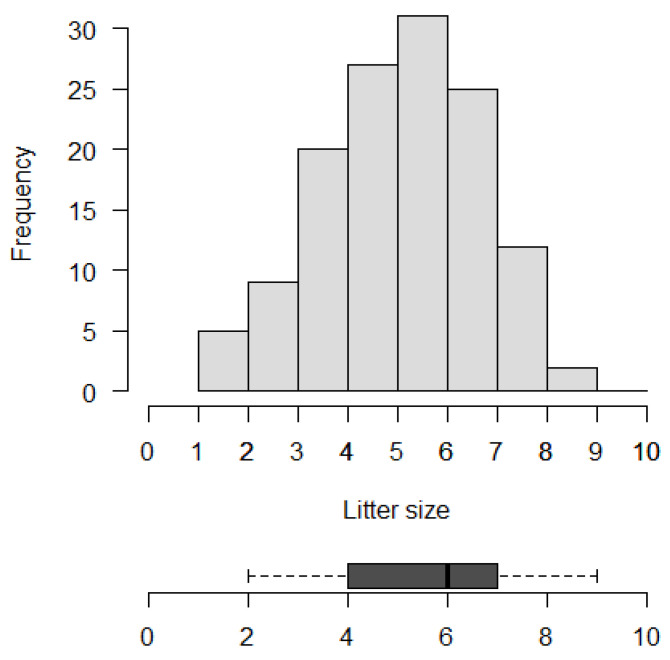
Frequency distribution histogram and box plot of litter size in the edible dormouse in Catalonia (n = 131). The vertical line in the box represents the median value, and the margins of the boxplot indicate the first and third quartiles.

**Figure 3 life-13-01223-f003:**
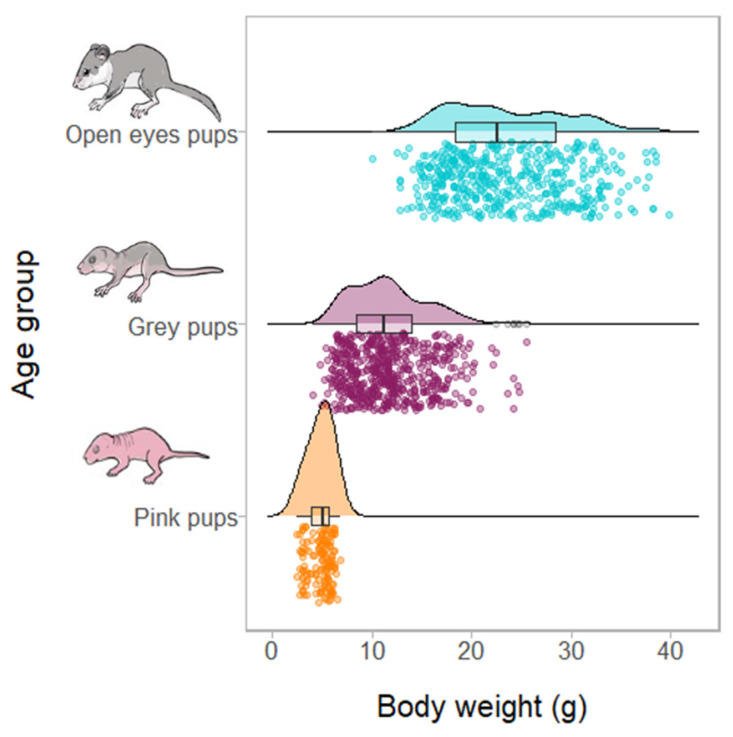
Density plots, box plots and data points summarizing the body weight of pups’ distribution in the edible dormouse according to age groups: pink, grey and open eyes pups.

**Figure 4 life-13-01223-f004:**
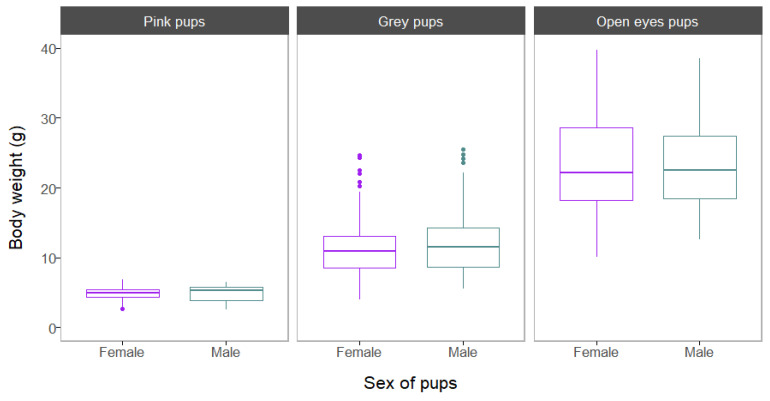
Box plots showing differences in offspring body weight between sexes in the three age groups considered.

**Figure 5 life-13-01223-f005:**
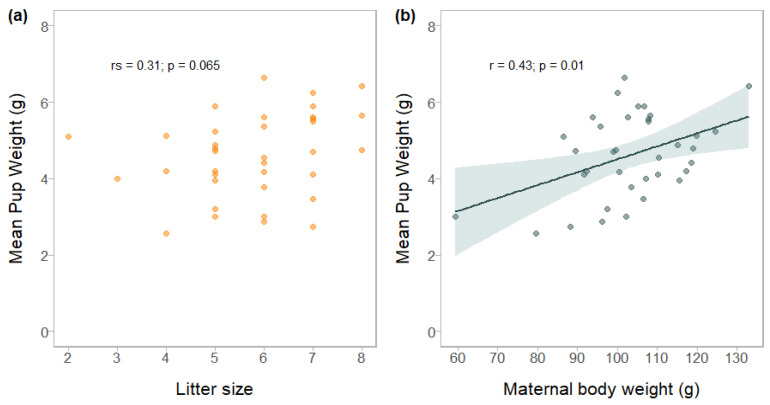
Scatterplots and simple linear regression model showing the relationship between (**a**) mean pup weight and litter size, and (**b**) mean pup weight and maternal body weight. The solid line indicates simple linear regression, the shaded band around the linear regression line represents 95% confidence intervals, and points refer to data. The correlation coefficients were calculated according to Pearson (r) or Spearman (r_s_), and a *p*-value lower than 0.05 was considered significant.

**Table 1 life-13-01223-t001:** Details of sampling stations in each monitored natural area according to sampling method, number of stations, sampling frequency, sampling period and years of study.

Natural Area Location	Sampling Method	Number of Stations	Sampling Frequency	Sampling Period	Year(s)
Montnegre i Corredor Park	Line transect	10	Two per year	Sept.; Oct.	2004–2011, 2020, 2021
	Plot	3	Biweekly	July–Nov.	2012–2021
Montseny Natural Park-Biosphere Reserve	Line transect	4	Two per year	Sept.; Oct.	2007–2011
	Plot	3	Biweekly	July–Nov.	2012–2021
Guilleries-Savassona Natural Area	Line transect	6	Two per year	Sept.; Oct.	2008–2014
	Plot	2	Biweekly	July–Nov.	2016–2021
Capçaleres del Ter i del Freser Natural Park	Plot	2	Monthly	July–Oct.	2017–2021
Vall d’Aran	Line transect	5	Monthly	July–Oct	2013–2021
Cadí-Moixeró Natural Park	Line transect	1	Monthly	July–Oct.	2019–2021
Andorra	Line transect	7	Two per year	July; Sept./Oct.	2008–2017

**Table 2 life-13-01223-t002:** Summary of model selection according to Akaike’s Information Criterion corrected for small sample size (AICc) for linear regression models and Generalized Linear Mixed Models (GLMMs) explaining variations in mean pup weight and litter size of the edible dormouse (*Glis glis*) in the NE of the Iberian Peninsula. The models are ranked according to their AICc in descending order of support.

Model Type	Model Formula	df	AICc	ΔAICc	*w_i_*
**Simple linear regression**	**mPW ~ MBW**	**3**	**106.8**	**0.00**	**0.566**
**Multiple linear regression**	**mPW ~ LS + MBW**	**4**	**107.9**	**1.14**	**0.321**
Simple linear regression	mPW ~ LS	3	111.2	4.46	0.061
Simple linear regression	mPW ~ 1	2	111.6	4.78	0.052
**Generalized Poisson GLMM**	**LS ~ PC1**	**7**	**524.6**	**0.00**	**0.916**
Conway-Maxwell-Poisson GLMM	LS ~ PC1	7	529.4	4.79	0.084
Poisson GLMM	LS ~ PC1	6	556.8	32.16	0.000

Abbreviations: mPW = mean pup weight; LS = litter size; MBW = maternal body weight; PC1 = first principal component selected from PCA that includes information related to geographic variables; df = degrees of freedom; AICc = Akaike Information Criterion value corrected for small sample sizes; ΔAICc = Delta AICc; *w_i_* = Akaike’s weight; ~1, the intercept-only model (null model). Note: Lower values of AICc indicate better model fit. Models in bold with ΔAICc < 2 have the greatest support in the data. Interactions between litter size and maternal body weight were not significant (Appendix A), and only models without interaction are shown.

**Table 3 life-13-01223-t003:** Results of the final linear regression model and Generalized Linear Mixed Model (GLMM) examining factors that affect mean pup weight and litter size of the edible dormouse (*Glis glis*) in the NE of the Iberian Peninsula, respectively.

Response, Model Formula, Model Parameter	Estimate	Standard Error	*p*-Value
**Mean pup weight:**			
mPW ~ LS + MBW *			
(Intercept)	0.943	1.331	0.4940
LS	0.052	0.102	0.6186
MBW	**0.033**	**0.013**	**0.0122**
**Litter size:**			
LS ~ PC1			
(Intercept)	1.716	0.024	<0.001
PC1	0.021	0.016	0.172

Abbreviations: mPW = mean pup weight; LS = litter size; MBW = maternal body weight; PC1 = scores of the first principal component of PCA, which includes information related to geographic variables. *P*-values considered significant (*p* < 0.05) are shown in bold. Mean pup weight model: R^2^_adj_ = 0.156–0.164, *p* < 0.05, n = 36; Litter size model; R^2^_m_ = 0.013, n = 135). Note: The table shows the best-selected models (see Section 2) testing the effects of litter size and maternal body weight on mean pup weight through linear regressions, as well as the effect of first principal component (PC1) scores on litter size through a Generalized Linear Mixed Model (GLMM) fitted using the generalized Poisson distribution and log link function, and including ’Sampling station’, ‘Nest box identity’, ‘Female identity’, and ‘Year’ as random effects. Results of random effects are omitted.* Averaging model.

## Data Availability

The data presented in this study are available on request from the corresponding author.

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
