# Peer review of "Influences of Maternal Weight and Geographic Factors on Offspring Traits of the Edible Dormouse in the NE of the Iberian Peninsula"

_life, 2023, doi:10.3390/life13051223_

Round 1

Reviewer 1 Report

English needs proof-reading. Grammar and lexical mistakes are numerous. 

Author Response

Response to Reviewer #1 Comments:

General Comment: Reproduction rate is the crucial characteristics for preserving endangered species and forecasting of its future abundance. Study of general laws of reproductive regulation is needed from theoretical and conservational points of view. The reviewed paper is a result of extensive monitoring program in especially protected nature territories, the obtained data from which can both contribute to the development of life history theory and become a basis for nature management decisions. The large volume of data is analyzed from several places and deep statistical analyses are conducted.

The authors collected the data on the litter size of the edible dormouse from different plots of Spain and Andorra, allowing to analyse trade-offs in the species reproduction and geographic variability. Several factors influencing on litter size and body mass of juveniles were studied. Results are given using GLM models and PCA analyses. The found regularities are discussed from the standpoints of life cycles theory and the concept of anticipatory reproduction.

Despite very careful descriptions of all procedures and results, I have some notes.

Response: Thank you very much. We appreciate the positive feedback from the reviewer. We have read your comments carefully and tried our best to address them one by one to improve after this revision. We hope that the manuscript changes now are follow your suggestions.

Comment 1: Introduction: Too many different theoretic concepts in a row.  It would be better to construct a separate paragraph on trade-offs and cost of reproduction (lines 32-46), then about maternal effects (lines 47-61) and finally about environmental factors (lines 61-74). It is rough to say that maternal effects are costs of reproduction, may be, one of consequences of trade-offs or complex of genetic and environmental factors influencing the posterity. Please, find the proper definition with references. 

Response: Thanks for your appreciate comments.

(1) We separated the text in three paragraphs following your comment: (a) trade-offs and cost of reproduction (lines 36-50 in the revised manuscript), (b) maternal effects (lines 51-66 in the revised manuscript) and (c) environmental factors (lines 67-80 in the revised manuscript).

(2)  We are really sorry for this error. The authors have changed this sentence from “One of the most important are maternal effects, that can be defined as the causal influence of genotypes or phenotypes of mothers on traits expressed by their offspring [15,16], impacting on survival and fitness of her progeny to modulate costs of reproduction, and therefore, playing an important role in life-history evolution, and constituting important adaptative mechanisms to environmental stress and changing environment [17-20]” to “Maternal effects can be defined as the causal influence of genotypes or phenotypes of mothers on the traits expressed by their offspring [15,16] and can impact offspring fitness, being important as adaptations to environmental stress and changing environment [17-20]”. Please refer to lines 52-58 on page 2 in the revised manuscript.

Comment 2: Lines 76-77: The distributional range of the edible dormouse embraces most of Europe and small Asian plot.

Response: We thank the referee for this suggestion, which led us to revise the explanations with regard to the distributional range of the edible dormouse, modifying the sentence and adding a new reference: ["Vekhnik, V.A. (2023). Distribution and habitats of the edible dormouse (Glis glis L., 1766), Journal of Wildlife and Biodiversity, 7(1), 13-39. DOI: https://doi.org/10.5281/zenodo.7675292"]. Therefore, according to the definition of the distributional range that appears in Vekhnik (2023), we have changed "…..and inhabits deciduous forests in western, central, and southeastern Europe [46,47]." to "…..and inhabits deciduous forests with a distributional range covering most of Europe, the Caucasus and Asia [50,51,52].". Please, refer to the revised manuscript (lines 89-91 on page 2). Thank you!

Comment 3: Line 122: Body mass can’t be strictly a maternal effect. It is only one indicator of physical condition of females. Influence of high or low female body mass on litter size, for example, may be one of manifestations of maternal effects.

Response: We apologize for unclear descriptions and thank you for your suggestion. Changes have been made as suggested. The sentence “whether body weight of the mothers is a maternal effect with relevant influence on offspring mass and litter size evaluated for the offspring age more nearest at birth“ have been modified as “whether maternal body weight has an effect on offspring weight and litter size using the pups nearest in age to birth". Please refer to lines 141-143 on page 3 in the revised manuscript. In addition, we also revised text and we have changed the sentence that appears in lines 745-746 on page 19 from "......understanding of maternal effect like maternal body weight......" to "......understanding of the effect of maternal body weight......" Now clarifications have been made in the revised manuscript.

Comment 4: Results: Lines 316-320: text repeat the data from the Table 2 and Figure 3. Currently, means and SD are given three times. The same for the Table 3 and Figure 4.

Response: Thank you very much for pointing this out. We are sorry for these repetitions. We revised the expression of section 3. Results. To reduce redundant information, we moved some redundant tables and figures to Supplementary Material (Appendix A) and only kept few example figures in the manuscript. Now, "Table A1" have changed to "Table A3" (line 249 on page 6). "Table 2" have moved to Appendix A where appears as "Table A1". "Figure 3" has been modified: “section (b)” has been moved to Appendix A where represents the "Figure A1", and we have changed Figure 3 caption from "Figure 3. Data visualization of body weight for the three offspring age groups in the edible dormouse: pink, grey and open eyes pups. (a) Density plots, boxplots and data points (b) Mean ± SD body weight of pups against age." to "Figure 3. Density plots, boxplots and data points summarizing the body weight of pups distribution in the edible dormouse according to age groups: pink, grey and open eyes pups" (lines 367-370 on page 10. Moreover, we also have moved "Table 3" to Supplementary Materials (Appendix A, now "Table A2" in the revised Appendices) to cut down the repetitions found. We have changed Figure 4 caption from "Figure 4. Box plots showing the differences in body weight between female and male in the three age groups of pups. Comparison using Mann-Whitney test for independent samples showed that females and males not differed in weight in any age group" to "Figure 4. Box plots showing  differences in offspring body weight between sexes in the three age groups considered" (lines 385-387 on page 11). Please refer to the revised manuscript and appendices.

Comment 5: Table 4: Probably, MBW instead of MPW?

Response: Thank you so much for catching this glaring and confusing error. Effectively, it is an error of the authors. Now, “MPW” has been corrected on “MBW”. Please refer Table 2 on page 12 in the revised manuscript.

Comment 6: Discussion: The detailed data on juveniles should be omitted or shifted to Results.

Response: Thank you for your comment. We agree to follow the reviewer's suggestion. Accordingly, we have removed the following sentences from the Discussion that repeat data appearing in the Results section. Concretely, we have eliminated the sentence "….collectively accounting for 63.4% (n = 83 litters)" (see line 481 on page 14 in the revised manuscript). We also have eliminated the paragraph "Concerning to offspring body weights, the overall mean weight in pink pups were 4.82 ± 1.13 g/pup and varied from 2.5-6.8 g (n = 140). In grey pups was observed a mean of 11.66 ± 3.88 g/pup (range = 4.0-25.5 g; n = 579) and for open eyes pups, the average weight was 23.60 ± 6.16 g/pup and ranged from 10.1 to 39.8 g (n = 451)" to the Discussion (lines 496-500 on page 14). In accordance with the previous change, we also revised and changed the text of the subsequent sentences appearing on lines 500-507 on page 14. Please refer to the revised manuscript.

Comment 7: Please, check this part for repetitions (for example, lines 460 and 479-480).

Response: Thank you very much for pointing this out. We revised these repetitions and we have modified the sentence as follows: "......... and (2) to evaluate the effect of maternal weight (used as a proxy of maternal condition or nutritional quality) on litter size and offspring weight in pink pups (age group nearest to birth) ". Please refer to lines 535-538 on page 15 in the revised manuscript.

Comment 8: Comments on the Quality of English Language: English needs proof-reading. Grammar and lexical mistakes are numerous. English proof-reading is needed.

Response: Thank you! We are very sorry for our incorrect writing. We revised the entire Manuscript and Appendices to eliminate grammatical and lexical mistakes. Numerous changes have been made to the revised manuscript and revised appendices, which are highlighted in red and embedded in the text. However, the requested extension was not granted, and we did not have time to submit the article for correction.

We also add a word.

Reviewer 2 Report

Reproductive patterns of the edible dormouse (Glis glis) populations in the NE of the Iberian Peninsula using an 18-year period of nest box monitoring data are analysed in the present manuscript. All parts of the manuscript are written in great detail, especially discussion is very thorough. Figures are well prepared and informative. References cited are relevant to the research, and the reference list is prepared very carefully.

However, the text of the manuscript is too long and should be more concentrated. There many repetitions. The same results are presented in the section 3. Results and repeated in the section 4. Discussion (e.g., lines 422-424, 430-433). Some results presented in Figures are repeated in the text (e.g., lines 304-307).

Title of the paper should be shortened. Too many key words, and some of them repeat the words of the title.

Some methodological concerns:

Pups were weighed with precision of 0.1 g, but all means are presented with precision of 0.01 g. The mean body weight in three age groups depends on the age of pups weighed: if more older pups would be weighed the mean body weight would be higher in the age group and vice versa. It is evident from the Figure 3, that values of body weight are widely scattered in grey pups and open eyes pups.

Analysis of possible geographic variation in litter size along elevational gradient (altitudes ranging from 460 to 1,497 m a.s.l.) is justified, but analysis of latitudinal or longitudinal gradient (distance just of few tens of kilometres) has no sense. The last analysis could be done across the entire Europe.

Influence of low-quality food availability in spring and summer can be discussed in Discussion section, but not presented in Abstract, because food availability and quality were not evaluated and not analysed in the present study.

Author Response

Response to Reviewer #2:

General Comment: Reproductive patterns of the edible dormouse (Glis glis) populations in the NE of the Iberian Peninsula using an 18-year period of nest box monitoring data are analysed in the present manuscript. All parts of the manuscript are written in great detail, especially discussion is very thorough. Figures are well prepared and informative. References cited are relevant to the research, and the reference list is prepared very carefully.

However, the text of the manuscript is too long and should be more concentrated. There many repetitions.

Response: Thank you very much for your previous comments that motived and helped us improve this manuscript, and for reminding us how important it is to present results in a more concise way.

Comment 1: The same results are presented in the section 3. Results and repeated in the section 4. Discussion (e.g., lines 422-424, 430-433). Some results presented in Figures are repeated in the text (e.g., lines 304-307).

Response: Thank you for pointing this out.

(1) We agree and therefore, we removed some repetitions and also restructured the sentences as suggested by the reviewer. Specifically, we have eliminated the sentence "…..collectively accounting for 63.4% (n = 83 litters)" (see line 481 on page 14 in the revised manuscript). We also have eliminated the paragraph "Concerning to offspring body weights, the overall mean weight in pink pups were 4.82 ± 1.13 g/pup and varied from 2.5-6.8 g (n = 140). In grey pups was observed a mean of 11.66 ± 3.88 g/pup (range = 4.0-25.5 g; n = 579) and for open eyes pups, the average weight was 23.60 ± 6.16 g/pup and ranged from 10.1 to 39.8 g (n = 451)" to the Discussion (lines 496-500 on page 14). In accordance with the previous change, we also revised and changed the text of the subsequent sentences appearing on lines 500-507 on page 14. Please refer to the revised manuscript.

 (2) Following the second comment, changes have been made as suggested. “Mean = 5.55; SD = 1.60; N =131; and the dashed red line indicating mean litter size inserted in Figure 2“ have been eliminated of the Figure 2 (please refer to Figure 2 on page 9 in the revised manuscript). Now, these results only appear in the text (lines 344-345). Consequently, the sentence “The mean, standard deviation (SD) and sampling size (N) are showed at top right. Dashed red line indicate the mean litter size of the studied population” in the descriptive caption of Figure 2 has also been eliminated (lines 356-357). However, we would like to keep the modified Figure 2 in order to facilitate the visualization of some results of the text if it is possible to include it. In addition, to reduce redundant information, we moved other tables and figures with repeated information to the Supplementary Material (Appendix A) and kept only a limited number of figures in the manuscript. Accordingly, "Table A1" have changed to "Table A3" (line 249 on page 6). "Table 2" have moved to Appendix A where appears as "Table A1". "Figure 3" has been modified: “section (b)” has been moved to Appendix A where represents the "Figure A1", and we have changed Figure 3 caption from "Figure 3. Data visualization of body weight for the three offspring age groups in the edible dormouse: pink, grey and open eyes pups. (a) Density plots, boxplots and data points (b) Mean ± SD body weight of pups against age." to "Figure 3. Density plots, boxplots and data points summarizing the body weight of pups distribution in the edible dormouse according to age groups: pink, grey and open eyes pups" (lines 367-370 on page 10. Moreover, we also have moved "Table 3" to Supplementary Materials (Appendix A, now "Table A2" in the revised Appendices) to cut down the repetitions found. We have changed Figure 4 caption from "Figure 4. Box plots showing the differences in body weight between female and male in the three age groups of pups. Comparison using Mann-Whitney test for independent samples showed that females and males not differed in weight in any age group" to "Figure 4. Box plots showing  differences in offspring body weight between sexes in the three age groups considered" (lines 385-387 on page 11). Please refer to the revised manuscript and appendices.

Comment 2: Title of the paper should be shortened. Too many key words, and some of them repeat the words of the title.

Response: Thank you for pointing out the inappropriate title and key words. We agree and have revised it as suggested by the referee.

(1) Title “Offspring number and size trade-off, maternal body weight effects and geographical patterns on litter size of the edible dormouse (Glis glis) populations in the NE of the Iberian Peninsula” has been changed to "Influences of maternal weight and geographic factors on offspring traits of the edible dormouse in the NE of the Iberian Peninsula". Please refer to lines 5-7 on page 1 in the revised manuscript.

(2) The initial key words “Glis glis; NE of the Iberian Peninsula; life-history theory; reproduction; trade-off; maternal effects; maternal body weight; litter size; offspring body weight; geographical gradient” has been reduced and modified to “Glis glis; Catalonia; reproduction; offspring number and size trade-off; maternal effects; litter size; offspring body weight; geographical gradients”. Please refer to lines 31-33 on page 1 in the revised manuscript.

Some methodological concerns:

Comment 3: Pups were weighed with precision of 0.1 g, but all means are presented with precision of 0.01 g. The mean body weight in three age groups depends on the age of pups weighed: if more older pups would be weighed the mean body weight would be higher in the age group and vice versa. It is evident from the Figure 3, that values of body weight are widely scattered in grey pups and open eyes pups.

Response: Thank you for your comment.

  • Yes, the digital balance used to obtain the weight provided the values to one decimal place. Nonetheless, when we calculated the averages some of the values gave more decimal places, and we decided to express the results in the paper with two decimal places. Ex., the mean value of 1.50 and 1.60 is 1.55, so a measure with 0.1 precision can yield mean values with more precision (0.01) and two decimal places. However, we agree with the reviewer to change the averages to express them to an accuracy of 0.1 g. Accordingly, we have expressed the averages to one decimal place. Please refer to lines 16-18 page 1, line 344 on page 8, lines 361-363 on page 9, lines 375-380 on page 11, line 398 on  page 12, line 480 on page 14, line 514 on page 15 and line 709 on page 18 in the revised manuscript.
  • We agree with the reviewer. The objective of this analysis was to obtain the mean pup weight and range in each of the three groups of pups according to literature, in order to describe the edible dormouse population present in Catalonia. We agree with the reviewer's comment that the weights of gray pups and open-eyed pups are more dispersed (as showed by Fig.3) . In this regard, we consider the greater dispersion of the data at these two ages to be logical and there can be reasons to explain this. One of the reason why the weight distribution is more homogeneous in the pink pups and more heterogeneous in the other two age groups may be due to the fact that the three ages differ in the range of days in which they remain in each group, but it is also important to consider that as they grow, weight differences within a litter and between litters become larger. In addition, maternal body weight account for a substantial proportion of the variation observed in offspring body weight, as discussed in an important part of this role. The ability of mothers to distribute resources evenly among their offspring may be limited by food resources and reduced by physiological constraints. Thus, variation in offspring weight within a litter could be due to constraints on maternal ability to allocate resources evenly. However, variation in offspring weight may not only be due to environmental or physiological constraints but may also be a strategy called diversified bed-hedging (Slatkin 1974, Seger and Brockmann 1987). This strategy consists of maximizing a number of offspring of variable weights with the aim to maximize fitness when environments vary temporally (Slatkin 1974, Seger and Brockmann, 1987). This strategy confers a reproductive advantage, since the offspring of a litter have different weights, which would be optimal for different environments. Although it is true that depending on the environment, some of these offspring would have a much lower probability of survival, the strategy also greatly reduces the possibility of losing all offspring as could occur if the offspring in a litter do not have variability in weight. In summary, animals sacrifice some of their potential fitness in order to reduce their probabilities of complete failure (Seger and Brockmann, 1987). Therefore, we believe that the variability observed between age groups is normal and justified, both because females can produce offspring of variable sizes (diversified bet-hedging) and because of the influence that food availability and physiological constraints can have, among other causes. We have also added some of this clarifications in lines 516-525 on page 15 in the revised manuscript. However, we remain at your disposition for any other comments you may have on this matter. Thank you very much!

References:

Slatkin, M. (1974). Hedging ones evolutionary bets. Nature, 250, 704–705.

Seger, J., & Brockmann, J. (1987). Oxford surveys in evolutionary biology, volume 4, chapter What is bet-hedging? (pp. 182–211). Oxford University Press.

Comment 4: Analysis of possible geographic variation in litter size along elevational gradient (altitudes ranging from 460 to 1,497 m a.s.l.) is justified, but analysis of latitudinal or longitudinal gradient (distance just of few tens of kilometres) has no sense. The last analysis could be done across the entire Europe.

Response: Thank you for this excellent observation. We are grateful for this comment, as it points to an important rationale of this study. The authors fully agree with the suggestion that a latitudinal/longitudinal analysis of litter size across Europe would be better, but this is out of the scope of this article. In this paper we showed original litter size data collected for over two decades in Catalonia and Andorra, and an analysis over the Entire European range will be more appropriate for a “review” paper. . We also agree with the comment that the available latitudinal and longitudinal gradients are small when we talk about degrees, and that altitudinal gradients are among the most powerful ‘natural experiments’ for testing ecological and evolutionary responses (Körner, 2007). In addition, environmental gradients can be easier to study on altitudinal gradients than on latitudinal gradients because a much greater range of conditions can be encountered over a much shorter distance (Dunn et al., 2009). However, some reproductive components respond differently to the latitudinal and altitudinal gradients (Daco et al., 2021), and the use of principal component analysis (PCA) allowed to control for the shared variance of latitude and altitude in the statistical models. Although the latitudinal and longitudinal gradient may be small in degrees, there is a larger gradient in minutes, and we think that possibly other local or regional peculiarities that we do not know could be captured despite the reduced range. Some studies on small mammals were carried out with more restricted latitudinal and longitudinal ranges than the one presented in this paper because the topography and climatology of those study areas varied markedly, and they even found that latitude and longitude did not show the same relationships with the distribution of small mammals (Moreno and Barbosa, 1992; Torre et al., 1996). Our study area also presents an orography that varies markedly. We believe that maintaining latitude and longitude could help us capturing some of this geographic variation that could also influence in the life-history trait evaluated. We also think that their inclusion could be justified because we are using the scores of the first principal component (PC) as an explanatory variable, representing as a whole the three original predictors (latitude, longitude and elevation).

In accordance with the referee's comment, we modified some sections of this paper. Concretely, we changed de following sentences: from “………within a strong gradient from the southernmost populations of the Iberian Peninsula located in Catalonia to the Pyrenees region in Andorra” to “………within a gradient from the southernmost populations of the Iberian Peninsula located in Catalonia to the Pyrenees region in Andorra” (line 145 on page 3); from “……..suggesting that number of pups per litter not followed any geographical trend neither had any effect of weather variation associated with the latitudinal and altitudinal gradients” to “……..suggesting that the geographic variation assessed (and climatic variation associated with the latitudinal and altitudinal gradients) does not affect litter size” (lines 467-470 on page 14); from “……..in litter size along a strong geographic gradient….” to “….in litter size along a geographic gradient……..” (line 731 on page 18). We have also added the following sentence: “Although some studies used more restricted latitudinal and longitudinal ranges than those used in the present study when the topography and climatology of the sampling areas varied markedly (Moreno and Barbosa, 1992; Torre et al., 1996), it would also be of particular interest to assess variation in litter size along a larger latitudinal and longitudinal gradients” (lines 698-701 on page 18). Please refer to the revised manuscript.

References:

Daco, L., Colling, G., & Matthies, D. (2021). Altitude and latitude have different effects on population characteristics of the widespread plant Anthyllis vulneraria. Oecologia, 197(2), 537–549. https://doi.org/10.1007/s00442-021-05030-6

Dunn, Robert R. and others, 'Chapter 3 Geographic Gradients', in Lori Lach, Catherine Parr, and Kirsti Abbott (eds), Ant Ecology (Oxford, 2009; online edn, Oxford Academic, 1 Feb. 2010), https://doi.org/10.1093/acprof:oso/9780199544639.003.0003

Körner C. (2007). The use of 'altitude' in ecological research. Trends in ecology & evolution, 22(11), 569–574. https://doi.org/10.1016/j.tree.2007.09.006

Moreno, E.; Barbosa, A. (1992). Distribution patterns of small mammal fauna along gradients of latitude and altitude in Northern Spain. Zeitschrift für Säugetierkunde. 1992, Vol 57, Num 3, pp 169-175

Torre, I., Tella, J.L., Arrizabalaga, A. (1996). Environmental and geographic factors affecting the distribution of small mammals in an isolated Mediterranean mountain. Zeitschrift fuer Saeugetierkd. 61, 365–375.

Comment 5: Influence of low-quality food availability in spring and summer can be discussed in Discussion section, but not presented in Abstract, because food availability and quality were not evaluated and not analysed in the present study.

Response: We think this is an excellent suggestion and we agree with your assessment. Accordingly, we have eliminated  the comments about the influence of low-quality food availability in spring and summer in the Abstract, keeping them in the Discussion. Concretely, the sentence "These findings could be suggesting that low-quality food availability in spring and summer could have an influence on maternal nutritional status and/or body mass during mating and gestation periods, affecting offspring weight at birth but not litter size" has been eliminated from the Abstract. Please refer to lines 22-24 on page 1 in the revised manuscript. Thank you!

We also add a Word.

Reviewer 3 Report

Dear Authors

I read your paper. There are some references that I strongly recommend you use and cite in your paper. One of these studies talks about edible dormouse mean litter size and nipple counts at the easternmost part of its global distribution, Iran's northern forests. It says that the mean litter size is around 7.5 and the mean females' weight is more than 277 gr. Therefore this is very important to discuss it. The title of the mentioned paper is as follows: Mammary number and litter size of the fat dormouse in the Southern Caspian coast

There is another review paper which I strongly recommend you to use it title is as follows:

       A Review on the Edible dormouse reproduction (Glis glis Linnaeus, 1766). 

Vekhnik, V. A., Ruf, T., & Bieber, C. (2022). A Review on the Edible dormouse reproduction (Glis glis Linnaeus, 1766). Journal of Wildlife and Biodiversity, 6(Special issue), 24–45. https://doi.org/10.5281/zenodo.7338112

The third paper which I recommend you to use and would like to see in the next review round is as follows

Moemen Beitollahi, S. ., Rajabi-Maham, H. ., Rezaei, H. R. ., & Naderi, M. (2022). Some ecological peculiarities of the Fat Dormouse (Glis glis Linnaeus, 1766) in Hyrcanian relict forests of Northern Iran. Journal of Wildlife and Biodiversity, 6(Special issue), 46–53. https://doi.org/10.5281/zenodo.7689765

I also have some questions:

Did you count the nipple variation? Is there any relationship between nipple count and the mean litter size?

Did you record the years where no mast yield? How the species matched with the mast yield? If they skipped the reproduction in the same years or it happens with a time lag?

My other comments have been embedded in the text/

Dear Authors

I read your paper. There are some references that I strongly recommend you use and cite in your paper. One of these studies talks about edible dormouse mean litter size and nipple counts at the easternmost part of its global distribution, Iran's northern forests. It says that the mean litter size is around 7.5 and the mean females' weight is more than 277 gr. Therefore this is very important to discuss it. The title of the mentioned paper is as follows: Mammary number and litter size of the fat dormouse in the Southern Caspian coast

There is another review paper which I strongly recommend you to use it title is as follows:

       A Review on the Edible dormouse reproduction (Glis glis Linnaeus, 1766). 

Vekhnik, V. A., Ruf, T., & Bieber, C. (2022). A Review on the Edible dormouse reproduction (Glis glis Linnaeus, 1766). Journal of Wildlife and Biodiversity6(Special issue), 24–45. https://doi.org/10.5281/zenodo.7338112

The third paper which I recommend you to use and would like to see in the next review round is as follows

Moemen Beitollahi, S. ., Rajabi-Maham, H. ., Rezaei, H. R. ., & Naderi, M. (2022). Some ecological peculiarities of the Fat Dormouse (Glis glis Linnaeus, 1766) in Hyrcanian relict forests of Northern Iran. Journal of Wildlife and Biodiversity6(Special issue), 46–53. https://doi.org/10.5281/zenodo.7689765

I also have some questions:

Did you count the nipple variation? Is there any relationship between nipple count and the mean litter size?

Did you record the years where no mast yield? How the species matched with the mast yield? If they skipped the reproduction in the same years or it happens with a time lag?

My other comments have been embedded in the text/

Author Response

Response to Reviewer #3:

General Comment: Dear Authors. I read your paper. There are some references that I strongly recommend you use and cite in your paper. One of these studies talks about edible dormouse mean litter size and nipple counts at the easternmost part of its global distribution, Iran's northern forests. It says that the mean litter size is around 7.5 and the mean females' weight is more than 277 gr. Therefore this is very important to discuss it. The title of the mentioned paper is as follows: Mammary number and litter size of the fat dormouse in the Southern Caspian coast

There is another review paper which I strongly recommend you to use it title is as follows:

               A Review on the Edible dormouse reproduction (Glis glis Linnaeus, 1766). 

Vekhnik, V. A., Ruf, T., & Bieber, C. (2022). A Review on the Edible dormouse reproduction (Glis glis Linnaeus, 1766). Journal of Wildlife and Biodiversity, 6(Special issue), 24–45. https://doi.org/10.5281/zenodo.7338112

The third paper which I recommend you to use and would like to see in the next review round is as follows

Moemen Beitollahi, S. ., Rajabi-Maham, H. ., Rezaei, H. R. ., & Naderi, M. (2022). Some ecological peculiarities of the Fat Dormouse (Glis glis Linnaeus, 1766) in Hyrcanian relict forests of Northern Iran. Journal of Wildlife and Biodiversity, 6(Special issue), 46–53. https://doi.org/10.5281/zenodo.7689765

Response: Thank you for your thorough review. We found your comments extremely helpful and have revised them accordingly. The authors agree with the reviewer's suggestion that these references need to be added. We have therefore added the three new references: Vekhnik et al., 2022; Naderi et al., 2014 and Moemen et al., 2022 in the reference list: [62,97,126]. In the following comments we detail where we have cited these resources. We hope you agree that they have improved the paper.

I also have some questions:

Comment 1: Did you count the nipple variation? Is there any relationship between nipple count and the mean litter size?

Response: Thank you for this very interesting question. It would have been interesting to explore this aspect. However, in our study, it was not possible to count nipples because the animals were not captured in kill traps (such as in Naderi et al. 2014), but nest boxes were used, and conducting this research in alive individuals would have represented using dangerous immobilization techniques (e.g., sedation of the females during nipple counting). In the current study, after capturing dormice in nest boxes, all individuals were marked, weighted, sexed, aged and measured as rapidly as possible in accordance to ethical guidelines for animals to minimize all disturbances. During year 2014, we tried to count nipples with live and non-sedated animals, but it was very difficult to obtain reliable data due to the high mobility of the females during manipulation. We immediately declined to continue this exploration mainly because of the stress that we think it could cause.

Comment 2: Did you record the years where no mast yield? How the species matched with the mast yield? If they skipped the reproduction in the same years or it happens with a time lag?

Response: Thank you for this excellent observation, so important for the study of the edible dormouse. We are sorry, but we did not record mast yield and seed production in most of the studied localities and years, so we were unable to analyze litter size and food availability in the study area.

My other comments have been embedded in the text/

Comment 3: Can you please add some data on the nipple count variation in edible dormouse at the Iberian peninsula?

Response:  We appreciate the reviewer's suggestion concerning the nipple count. Even so, the authors would like to clarify that in this monitoring program we work with live animals, and it is not possible to obtain nipple count data. Furthermore, this process cannot guarantee ethical guidelines for animal management, as justified in comment 1. We are sorry, but we cannot add nipple count variation data of the edible dormouse at the Iberian Peninsula.

Comment 4: Keywords better not to repeated from the title words.

Response: We thank the referee for this suggestion, which helps us clarify our paper. We fully agree and accordingly, we changed keywords and title as following:

(1) Title “Offspring number and size trade-off, maternal body weight effects and geographical patterns on litter size of the edible dormouse (Glis glis) populations in the NE of the Iberian Peninsula” has been changed to "Influences of maternal weight and geographic factors on offspring traits of the edible dormouse in the NE of the Iberian Peninsula". Please refer to lines 5-7 on page 1 in the revised manuscript.

(2) The initial key words “Glis glis; NE of the Iberian Peninsula; life-history theory; reproduction; trade-off; maternal effects; maternal body weight; litter size; offspring body weight; geographical gradient” has been reduced and modified to “Glis glis; Catalonia; reproduction; offspring number and size trade-off; maternal effects; litter size; offspring body weight; geographical gradients”. Please refer to lines 31-33 on page 1 in the revised manuscript.

Comment 5: I would like to cite the following paper as well:

Vekhnik, V. A., Ruf, T., & Bieber, C. (2022). A Review on the Edible dormouse reproduction (Glis glis Linnaeus, 1766). Journal of Wildlife and Biodiversity, 6(Special issue), 24–45. https://doi.org/10.5281/zenodo.7338112

Moemen Beitollahi, S. ., Rajabi-Maham, H. ., Rezaei, H. R. ., & Naderi, M. (2022). Some ecological peculiarities of the Fat Dormouse (Glis glis Linnaeus, 1766) in Hyrcanian relict forests of Northern Iran. Journal of Wildlife and Biodiversity, 6(Special issue), 46–53. https://doi.org/10.5281/zenodo.7689765

Response: Thank you for these interesting references suggested by the referee. We understand the referee’s general point and have included additional sentences related to the references that the reviewer recommend citing in our paper. Concretely, we have added the suggested content as follows:

Vekhnik et al., 2022: These article was already cited in the initially submitted manuscript. However, we have added more information about this article to our paper. Please refer to lines 492-493 on page 14 and line 613 on page 16 in the revised manuscript. Thank you!

Moemen et al., 2022: We added the sentence “The results are also in consonance with the importance that fruits may have in spring and summer, since in other regions it was found that the presence of trees producing edible fruits such as different berries, plum trees, grapes, common figs or walnut affect edible dormouse habitat use [126]”. Please refer to lines 633-636 on page 17 in the revised manuscript.

Comment 6: Here you need to talk about the nipple variation among the mammals. I hope you already counted the nipples.

Response: Thank you for pointing this out.

  • We agree with the reviewer that it is important add a description of nipple variation among the mammals in the place of the Introduction indicated by the referee. We have added the suggested content to the manuscript as following: “The number of mammae and nipples varies greatly among groups of mammals, ranging from 2 to 29 [46]. Not only traits such as mother's weight, seasonality, resource availability or environmental features are correlated with the number of offspring per litter [11,32,47,48], but also mammae and nipples number can explain part of the variation in litter size [48,49]. Specifically, the number of mammae and nipples are usually positively correlated with litter size [49]. Rodent species have on the average one-half as many offspring as they have nipples [49]”. Please refer to lines 81-87 on page 2 in the revised manuscript. Thank you very much. We believe that the manuscript is now much better.

References added to the Manuscript:

[46] Feldhamer, G.A.; Merritt, J.F.; Krajewski, C., Rachlow, J.L.; Stewart, K.M. Mammalogy: Adaptation, Diversity, Ecology; Fifth edition.; Johns Hopkins University Press: Baltimore, 2020; ISBN 978-1-4214-3652-4.

[47] Battistella, T.; Cerezer, F.; Bubadué, J.; Melo, G.; Graipel, M.; Cáceres, N. Litter Size Variation in Didelphid Marsupials: Evidence of Phylogenetic Constraints and Adaptation. Biol. J. Linn. Soc. 2019, 126, 40–54. https://doi.org/10.1093/biolinnean/bly157

[48] Stewart, T.A.; Yoo, I.; Upham, N.S. The Coevolution of Mammae Number and Litter Size. bioRxiv 2020. https://doi.org/10.1101/2020.10.08.331983.  

[49] Gilbert, A.N. Mammary Number and Litter Size in Rodentia: the “One-Half Rule”. Proc. Natl. Acad. Sci. 1986, 83, 4828-4830. https://doi.org/10.1073/pnas.83.13.4828.

  • It would have been interesting to explore this aspect. Thank you. However, in our study, this is not possible because we work with live animals. Moreover, the aim of this study was analyzing possible relationships between maternal weight, litter size and offspring weight, taking a clear approach with maternal effects and life-history trade-offs. A second goal was to explore the geographic variations in litter size related with latitude, longitude, and elevation, using these predictors as a proxy of changes in environmental variables and therefore, under the focus of climatic variations approach.

Comment 7: Can you please provide a graph that shows the mast yield and its trend over your study period? Which years there were no mast production and how the target species responsed? Is reproduction skipped at the same year? or year after the non-mast yield year.

Response: Thank you very much for this important suggestion. According to the previous comment (#2), we have not information on seed production in almost all the localities and years covering that study.

Comment 8: Usually ear pad can be damage in this animals.

Response: Thank you very much for this appreciation. However, an important part of this study (which is not covered in the current paper) tries to know temporal variations of the populations, as well as population size in the study areas. Therefore, capture-mark-recapture method is necessary to accomplish these objectives. When animals are marked with ear tags, the process is carried out carefully, and it is rare to damage the ear if it is conducted by a qualified person. During the study period, many animals have been recaptured several times both within the same year and between years. We would like to clarify that the authors have carried out many field samplings every 15 days, and we have never observed infections and is very unusual to observe damaged ears. In addition, the authors would like to argue, based on our observations, that it is very unusual that dormice lose their ear tags. When juveniles are marked, the growth of the ear is also considered, leaving enough space to allow for growth. Furthermore, this method is fairly accepted in ethical guidelines (Sikes et al. 2016) and used by mammalogists working with small mammals worldwide.

Comment 9: Only females? or both genders? if yes why?

Response: Thank you for your question. We have specified in the paper that both sexes were marked. Please refer to line 214-215 on page 6 in the revised manuscript. We apologize for not specifying in the manuscript why both sexes are marked, but this would require explaining part of the monitoring that is not covered in this article and we cannot expand further with the text. We marked males and females of all ages (adults, yearlings and juveniles), but not pups. Some of the objectives were to estimate population size and survival rates and to use this information for effective management of the species.

Comment 10: Too long and hard to follow. Please rewrite.

Response: We apologize for unclear descriptions and thank you for your suggestions. To improve the clarify of the manuscript, this sentence has been modified as follows: “The present study analyzed the reproductive patterns of edible dormouse (Glis glis) populations in the northeast of the Iberian Peninsula (Spain and Andorra) using an 18-year period of data obtained from nest boxes collected between 2004 and 2021. The results provide the first overall description of litter size and the offspring weight in a large territory of Catalonia (Spain), and not only focusing the study on two specific natural parks as was carried out in a previous work [94]”. Please refer to lines 472-479 on page 14 in the revised manuscript. We hope that the paragraph will be much more understandable.

Comment 11: For Iran it was reported 7.85 ± 0.89. May be can be interesting to discuss, as the Iranian fat dormice is the largest one in the world. 

https://www.researchgate.net/publication/258927308_Mammary_number_and_litter_size_of_the_fat_dormouse_in_the_Southern_Caspian_coast

Response: Thank you for your suggestion. We agree with the reviewer's assessment. Accordingly, we have added the following sentences in the Discussion, citing the reference suggested by the reviewer: “ However, the value obtained was considerably lower than the 6.8 found in England [54] or litter size of 7.85 estimated in Iran [97]. Litter size can vary depending on the number of studied litters, forages conditions of a locality in certain years or mammary number [62,97], among other causes. In the particular case of the Iranian population, the large difference could also be due to the different methodology used, because individuals were captured with kill traps and litter size was estimated from embryos and placental scars [97].”. Please refer to lines 490-496 on page 14 in the revised manuscript.

We also add a Word.

Round 2

Reviewer 2 Report

The authors took into account all my comments and answered them in detail.

Reviewer 3 Report

Dear Authors

Thanks for applying the comments and suggestions.